# Effectiveness and Feasibility of Workplace-Based Mental Health Interventions for University Academic Staff: A Systematic Review

**DOI:** 10.3390/ijerph22121787

**Published:** 2025-11-26

**Authors:** Veena Abraham, Johanna C. Meyer, Kebogile Elizabeth Mokwena, Edward Duncan, Xuan Luu, Kathryn Hinsliff-Smith

**Affiliations:** 1Department of Pharmaceutical Sciences, School of Pharmacy, Sefako Makgatho Health Sciences University, Pretoria 0204, South Africa; 2Faculty of Health Sciences and Sport, University of Stirling, Stirling FK9 4LA, UK; edward.duncan@stir.ac.uk; 3Department of Public Health Pharmacy and Management, School of Pharmacy, Sefako Makgatho Health Sciences University, Pretoria 0204, South Africa; hannelie.meyer@smu.ac.za; 4South African Vaccination and Immunisation Centre, Sefako Makgatho Health Sciences University, Pretoria 0204, South Africa; 5Department of Public Health, School of Healthcare Sciences, Sefako Makgatho Health Sciences University, Pretoria 0204, South Africa; kebogile.mokwena@smu.ac.za; 6School of Social Sciences, College of Arts, Law and Education, University of Tasmania, Hobart, TAS 7005, Australia; xuan.luu@utas.edu.au; 7Leicester School of Nursing and Midwifery, Faculty of Health and Life Sciences, De Montfort University, Leicester LE1 9BH, UK; kathryn.hinsliff-smith@dmu.ac.uk

**Keywords:** workplace mental health, academia, burnout, stress, employee wellness, higher education, occupational health, workplace safety

## Abstract

Academic staff face workplace stressors such as high workloads, job insecurity, and limited institutional support, which contribute to psychological distress and burnout. While workplace-based interventions are important for maintaining well-being, their effectiveness in academic settings remains underexplored. This systematic review including qualitative, quantitative, and mixed methods studies synthesized evidence on individual-level mental health interventions for university academic staff. Five databases (PsycINFO, MEDLINE, Web of Science, SCOPUS, ERIC) were searched for peer-reviewed studies published between 2003 and 2023. From 1058 records, seven studies met the inclusion criteria. Methodological quality was appraised using the Mixed Methods Appraisal Tool (MMAT) and the GUIDance for the rEporting of intervention Development (GUIDED) framework. Interventions included lifestyle regimens, mindfulness, yoga, community therapy, and institutionally embedded wellness programs. The findings suggest that brief, structured, and theoretically grounded interventions can improve psychological well-being, reduce burnout, and enhance coping. Facilitators included leadership support, institutional integration, peer networks, and flexible delivery; barriers included stigma, workload pressures, attrition, and digital challenges. Most studies were conducted in the Global North, limiting transferability to resource-constrained contexts. Overall, individual-level interventions show promise, but sustainable, culturally adapted, and multilevel strategies are needed to strengthen mental health support in higher education. These insights also contribute to broader efforts to promote health and safety in the workplace by identifying practical strategies to enhance well-being across diverse occupational settings.

## 1. Introduction

Mental health has emerged as a global priority area with substantial implications for international policy and practice. The World Health Organization (WHO) has defined mental health as “a state of well-being in which the individual realizes his or her own abilities, can cope with the normal stresses of life, can work productively and fruitfully, and is able to make a contribution to his or her community.” [1] Alarmingly, 1 in 8 people are living with a mental disorder, resulting in over 970 million people affected globally [2]. Recent global evidence suggests that depression and anxiety lead to 12 billion lost working days annually, with associated productivity losses costing approximately USD 1 trillion per year [3,4].

Health promotion, and by extension, mental health promotion, refers to action and advocacy that address modifiable social determinants of health. This requires broad participation from stakeholders [5,6] while also including wide-ranging opportunities for implementation at various levels: societal, sub-population, and individual [7,8]. One of the tenets promoting positive mental health is the development and maintenance of healthy communities, which includes a positive working environment [9,10]. Burnout and work-related stress typically arise when job demands chronically exceed available resources, for example, excessive workloads, conflicting roles, unclear expectations, or limited support from leadership and peers. Such conditions undermine psychological safety and autonomy, leading to emotional exhaustion, depersonalization, and diminished occupational health [11,12,13].

Workplace characteristics and culture may contribute towards burnout, anxiety, and depression [3,14]. Interventions should equip employees with coping strategies or adjust the work environment. Performance management systems are currently geared towards high output, causing employees to resort to extreme measures to meet the demands, which may lead to unhealthy outcomes [15]. Maintaining mental health in the workplace impacts personal well-being, community functioning, and national economic productivity [7,16]. The goals of effective interventions often include improving role clarity and alleviating stress. These can be achieved through various approaches, such as organizational restructuring, staff development programs, or targeted mental health training [17]. Despite increased awareness, formal policies and implementation thereof remain a concern. An umbrella review [18] found that high-intensity workplace-based interventions and access to clinical management improve outcomes. While the WHO stresses the importance of multilevel strategies to prevent, protect, and promote mental health at work [3], interventions in practice have typically focused more on individuals with limited integration of holistic strategies involving both individual and organizational reforms [10]. A truly holistic approach should also account for physical and environmental factors (e.g., lighting, noise, air quality, temperature, and workspace ergonomics), which can significantly affect concentration, fatigue, and overall mental well-being [19,20,21,22].

Although universities share many characteristics with other workplaces, academic staff face distinct challenges such as research output demands, student expectations, and administrative workloads. Bridging evidence from general workplace initiatives to this context is therefore crucial to identify which approaches are most adaptable and effective in academia.

Within higher education settings, the mental health of academic staff is closely tied to outcomes such as stress, burnout, depression, and anxiety, which have far-reaching effects on both individual well-being and institutional performance. High levels of occupational stress and emotional exhaustion are associated with reduced productivity, job dissatisfaction, absenteeism, and turnover, compromising teaching quality and research output [13,23,24]. Conversely, improved psychological well-being has been linked to higher engagement, creativity, and organizational commitment [11,25]. Understanding how workplace interventions can influence these outcomes is therefore central to improving both staff health and university effectiveness. In the academic setting, the marketization of universities has increased occupational stress among academic staff, leading to higher levels of depression, burnout, and other mental health issues. This has been highlighted in the United States [26] as well as across diverse global contexts, including several low- and middle-income countries [27]. This phenomenon is largely driven by the impact of neoliberalism in higher education globally, a sociopolitical approach that typically places more emphasis on an individual’s responsibility towards performance with less focus on institutional support and collective bargaining power. This has also been observed in the South African academic context, where these dynamics have been shown to contribute to elevated stress and burnout among academic staff in this setting [28]. The implementation of performance-based funding/metrics, highly competitive academic environments, and the commodification of education have resulted in the exploitation of academic staff, further exacerbating stress and burnout in this population [27]. South African scholars have even described this impact as modern academic slavery [29]. The pandemic underscored the critical need for accessible mental health resources and support systems for academic staff [30,31] and compounded existing sectoral stressors such as job insecurity, institutional restructures, and performance-driven work cultures [32,33,34].

A systematic review on mental health promotion interventions at universities published in 2016 [35] highlighted a lack of evidence focusing on interventions targeting academics compared to students, with only four of the nineteen included studies focused on staff [36,37,38,39]. Another recent systematic review examined structural and organizational-level interventions in university settings, again finding limited attention to academic staff compared to students [40]. More recently, a scoping review identified stressors and interventions in higher education institutions [41], but was limited to studies published during the COVID-19 pandemic (2020–2023) and did not assess study quality. In a recent, similar systematic review [42], well-structured interventions that incorporated behavioral change elements together with clear delivery plans were found to be most effective among academic staff. Our review aimed to build on this evidence by exploring and assessing the quality of interventions specific to academic staff in universities, reporting on barriers and limitations within this context.

Conducting this systematic review on mental health promotion programs in universities is essential due to the unique challenges in these settings. Whilst existing studies have explored workplace mental health programs in various contexts, academic staff in universities face specific pressures related to workload and performance expectations, including heavy teaching loads, pressure to publish, competition for research funding, job insecurity, student demands, and administrative burden. Therefore, a dedicated review is crucial for identifying effective strategies tailored to the challenges faced by today’s academics. The majority of studies in higher education institutions (HEIs) have focused on student mental well-being, often overlooking academic staff [35]. This review highlights key barriers and facilitators for mental well-being interventions for academics and HEI policy makers.

Although research on workplace mental health has expanded considerably, there has historically been limited focus on interventions tailored to academic staff. Only recently have systematic reviews begun to examine this population specifically [40,42,43,44,45,46,47]. However, these reviews tend to address either specific subgroups, or narrow intervention types, and few have comprehensively synthesized evidence on both the effectiveness and implementation factors of interventions across diverse university settings.

This review builds on these emerging efforts by providing an updated and integrative synthesis of effectiveness, feasibility, and barriers/facilitators to implementing mental health and well-being interventions for university academic staff.

### 1.1. The Present Review: Purpose and Guiding Questions

A reliable evidence base is crucial for the development of future interventions; thus, this systematic review examined the range and nature of the evidence on workplace-based mental health interventions in academic settings; synthesized and disseminated the research results; and identified the literature gaps to guide intervention development.

A systematic review was considered the most suitable approach as it allows for rigorous quality appraisal, structured data synthesis, and clearer assessment of intervention effectiveness, steps that scoping or narrative reviews do not typically provide.

### 1.2. Specific Questions

What is the reported effectiveness of workplace-based mental well-being interventions implemented for academic staff in university settings?What are the reported key barriers and facilitators to implementing workplace-based mental well-being interventions in this sector?

## 2. Materials and Methods

The systematic review including qualitative, quantitative, and mixed methods studies was conducted following the 27-item checklist PRISMA 2020 statement, which guides reporting of a clear and accurate systematic review [48]. Refer to Appendix A for the completed checklist. The review protocol was registered with PROSPERO https://www.crd.york.ac.uk/prospero/display_record.php?ID=CRD42023446197 (accessed on 17 July 2023), registration number CRD42023446197.

### 2.1. Eligibility Criteria

#### 2.1.1. Types of Studies

The PICO framework, an abbreviated format outlining Participants, Intervention(s), Comparison(s), and Outcome(s) for the systematic review of interventional studies, guided the inclusion criteria for the systematic review and is elaborated on below.

#### 2.1.2. Participants/Population

For this review, eligible studies included those involving all academic staff in university settings; this included any personnel employed by an institution of higher learning who are involved in teaching and/or research activities. This excluded administrative, support, or technical staff not engaged in core academic functions. Studies that included both academic and administrative staff were only eligible if outcomes for academic staff were reported separately; however, no such mixed-group studies met our inclusion criteria.

#### 2.1.3. Intervention

A publication or article was eligible for inclusion in the review if it reported implementation strategies, facilitators, or barriers to mental health promotion interventions in academic spaces. While the review predominantly focused on interventions oriented towards individual-level action, programs operating at both individual and organizational levels (including those that describe intervention development) were included, where relevant. In this review, individual-level interventions were defined as those targeting personal behaviors, coping skills, or stress management capacities of academic staff (e.g., mindfulness training, resilience workshops, cognitive behavioral programs). Organizational-level interventions were defined as strategies directed at modifying structures, policies, leadership practices, workload management, or institutional culture to enhance mental well-being. This dual inclusion ensured that the synthesis captured both person-focused and system-level approaches relevant to the academic work environment.

#### 2.1.4. Comparison

We included studies with any comparison/control group that focused on intervention impact and outcomes. Studies without such groups were still eligible if they reported on implementation strategies, feasibility, acceptability, or barriers and facilitators relevant to mental health and well-being interventions.

#### 2.1.5. Outcome

This review included studies that assessed primary outcomes related to mental well-being, including psychological, emotional, and psychosocial health and well-being, wellness, self-efficacy, self-esteem, empowerment, quality of life, and flourishing. Secondary outcomes included common mental disorders (e.g., depression, anxiety, personality disorders) and burnout.

### 2.2. Study Design

Study designs included in the review were qualitative research, randomized controlled trials, non-randomized studies, quantitative descriptive studies, and mixed methods studies as categorized by the Mixed Methods Appraisal Tool (MMAT). Any original peer-reviewed research study (quantitative, qualitative, and mixed methods, and inclusive of systematic and scoping reviews) about workplace-based mental well-being interventions for academic staff (teaching and/or research roles) in university settings was included. Thus, the review included studies that were

Targeted at academic staff who were identified with teaching and research responsibilities;Located within a university setting (any geographic location);Published in English in peer-reviewed journals;Inclusive of at least one of the following outcomes of interest:Self-reported mental health or well-being outcomes (e.g., psychological distress, burnout, resilience, self-efficacy);Intervention planning, delivery, feasibility, acceptability, or identified barriers and facilitators to implementation.

We excluded the gray literature (including dissertations and thesis), editorials, commentaries, letters, clinical trial protocols, and narrative reviews.

### 2.3. Information Sources and Search Strategy

The search for relevant studies was conducted in the following five electronic databases: PsycINFO, MEDLINE, Web of Science, SCOPUS, and ERIC. The PROSPERO database was searched before commencement of the review to ensure no other similar reviews were published or underway. Searches were carried out between November 2023 and January 2024, limiting the date range of publication to between 1 January 2003 and 31 December 2023. This 20-year window was selected to capture contemporary developments in workplace and mental health promotion research, during which time the field matured conceptually and methodologically following the WHO’s 2001 and 2002 landmark workplace health frameworks and the global shift towards evidence-based and innovative intervention practices. Limiting the range ensured inclusion of studies that reflect current university work conditions, digital delivery modes, and post-2000 higher education reforms, while excluding the older literature less comparable to modern institutional contexts. The search was completed in January 2024 and incorporating studies published post-2023 was beyond the project’s resources and time constraints. Future reviews are encouraged to update these findings as new evidence emerges.

Iterative methods were used to develop the search strategy for this review. The initial search focused on key concepts related to academics, mental health interventions, university settings, and the effectiveness of these interventions. Through consultation with a university librarian, additional relevant terms were identified for inclusion in the search. Boolean operators (AND/OR) were used to combine terms and proximity searches were conducted to maximize the retrieval of relevant studies. The search strategy was adapted for each database with predefined search strings that included variations in university staff roles (e.g., faculty, lecturers, researchers, etc.) and mental well-being interventions. A detailed breakdown of search terms for each database is outlined in Appendix A.

#### 2.3.1. Study Selection and Screening

A PRISMA flow chart was utilized to track the number of studies at each stage of the review. All final searches were executed by the first author and exported into EndNote Reference Management (version 20) software where duplicates were automatically removed. These records were then imported to COVIDENCE systematic review software [48]. Two authors independently screened titles and abstracts on COVIDENCE. Subsequently, full texts of potentially eligible studies were also screened by two authors to further check for eligibility. All conflicts at different stages of screening were resolved by the two reviewers through discussions, leading to a consensus. A third reviewer was available to adjudicate unresolved conflict; however, no such cases arose. Additionally, the reference lists of all the included studies (*n* = 7) were searched and cross-referenced by one reviewer to supplement the search; however, no additional studies were added to the review from this backward citation search.

#### 2.3.2. Data Extraction

For each included study, two authors independently extracted data. This included study design, study aims, population characteristics (sample size, demographics), inclusion and exclusion criteria, intervention details (type and description, duration, delivery mode, setting, targeted outcomes), comparison group information and outcomes, primary and secondary outcomes, summary of quantitative and qualitative results, implementation factors (facilitators/barriers to implementation, stakeholder involvement, institutional support), and study conclusions, limitations, and recommendations. There were no conflicts at this stage of data extraction.

#### 2.3.3. Quality Appraisal

Two different appraisal tools were used to assess included studies, reflecting the differences in study type. The Mixed Methods Appraisal tool (MMAT [49]) was used to evaluate the methodological quality of studies that reported on the effectiveness of interventions. The MMAT allows for assessing qualitative, quantitative, and mixed methods designs, highlighting potential biases, participant attrition, and validity of findings. The MMAT assesses methodological quality using 5 criteria (appropriateness of study design, participant selection, data collection methods, risk of bias in measurement, and coherence between qualitative and quantitative aspects for mixed methods). Each study was appraised by answering each criterion with “Yes”, “No”, or “Can’t tell”. The reviewers discussed discrepancies, and a third reviewer was consulted in cases of disagreement.

Three of the seven included studies reported on the development of interventions instead of intervention implementation; thus, it was not appropriate to use the MMAT to appraise them. Instead, these studies were assessed using the GUIDED tool [50]. This tool provides a structured framework for the process of developing interventions in health research. Originally developed for the reporting of intervention development studies, its criteria also provide a useful structure for appraising the quality of studies. Each study was assessed against key GUIDED criteria, including context, development process, use of theory and evidence, intervention components, adaptability, feasibility, stakeholder engagement, and evaluations.

Two reviewers independently appraised each study, using the MMAT template or GUIDED tool, and compared appraisals to arrive at a consensus. Studies were not excluded based on critical appraisal.

#### 2.3.4. Data Synthesis

A meta-analysis was not feasible for this review due to the methodological diversity of included studies. To incorporate the different study designs (quantitative and qualitative) and appraisal types (MMAT appraisal and GUIDED evaluation), a dual evaluation and narrative synthesis approach was followed for this review. The narrative synthesis was conducted in line with established guidance [51,52,53], which recommends the development of preliminary synthesis, examining any relationships within and between studies and assessing the robustness of the synthesis.

Studies were categorized based on the intervention type reported in the included studies and analyzed thematically for effectiveness, feasibility, barriers, and facilitators. Quantitative findings were summarized descriptively while qualitative findings were examined for thematic patterns. Insights from MMAT and GUIDED appraisals were integrated to enable the consideration of methodological quality alongside the feasibility and practicalities of implementation. Several recent reviews have adopted a similar approach [54,55,56].

Tabular summaries are included to present key results such as study characteristics and methodological quality.

## 3. Results

### 3.1. Study Selection

The database searches yielded 1635 records. Following initial de-duplication in EndNote (538 duplicates identified and removed), 1097 records were imported into the online platform COVIDENCE [48], for subsequent management. COVIDENCE identified a further 36 duplicates, and an additional 3 duplicates were found manually by the reviewers, leaving 1058 records for title and abstract screening. Screening was conducted independently by two reviewers and resulted in 64 records being selected for full-text review, of which 7 met the inclusion criteria. All steps in the review process were conducted independently by two reviewers.

Backward citation searching of the 7 included studies was conducted by a single reviewer and yielded 315 additional records. Of these, 84 were excluded because they were published before 2003, and the remaining 231 were excluded because they did not meet the inclusion criteria. No additional studies were included from the backward search.

The study selection process is detailed in the PRISMA 2020 flow diagram below (Figure 1).

Study characteristics: 

Of the seven included studies, four assessed intervention effectiveness [57,58,59,60] and three focused on intervention development [61,62,63]. Regarding methodology, three were quantitative, one was qualitative, and three utilized mixed methods. All articles were published in peer-reviewed journals. Participant numbers ranged between 8 and 141 participants across most studies. Contrastingly, Innstrand and Christensen [62] reported the number of survey responses rather than individual participants, collecting over 15,000 responses across 18 universities and university colleges in Norway as part of their intervention planning process. This reflects a difference in the unit of analysis, with most studies focusing on direct participant counts and one capturing large-scale institutional input through survey data.

Only two studies [58,59] included a control group, while the rest were pre–post or descriptive evaluations. Six studies were conducted in the Global North with only one study (Sena et al., 2023) [60] from the Global South. Table 1 outlines the key characteristics of the included studies, while Table 2 provides a summary of the theoretical underpinnings of each intervention along with the mental health outcomes that were assessed in each study. This pairing in Table 2 highlights how different theoretical models align with targeted outcomes, supporting the interpretation of the intervention design choices and their potential applicability in other contexts.

#### 3.1.1. Methodological Quality and Intervention Development

MMAT appraisal:

As previously described, the MMAT assesses factors such as clarity, data appropriateness, and adherence. Among the four effectiveness-focused interventions, those that employed randomized or controlled designs [59] showed stronger methodological rigor while those with high attrition rates [58] and self-reported outcomes without triangulation from other data sources [60] had lower confidence levels. Table 3 summarizes the quality appraisal findings of these four included interventions.

The MMAT appraisal leads us to posit that the most effective interventions in these settings are the yoga-based intervention [59] and community-based therapy [60]. These interventions were well-structured and benefited from strong participant engagement. While the former [59] also received institutional backing, the latter [60] emerged from staff members’ collective efforts rather than formal support from the institution. Rigor was greatest in the yoga-based intervention study by Rodríguez-Jiménez et al. [59], which demonstrated reduction in bias through an experimental pre–post design, incorporation of randomization with controls, and use of multiple validated outcome measures.

The MMAT appraisal indicated strong methodological quality for the community-based therapy intervention [60]. This intervention showed strong psychological benefits; the authors documented the qualitative methodology robustly, and the theoretical underpinnings grounding the intervention were also sound. The only limitation for this intervention is that of a small sample size, which limits the generalizability of their findings.

The NEWSTART program [57] was assessed as moderately effective; this intervention had beneficial effects for the measured outcomes (self-efficacy and stress management); however, concerns remain over its long-term sustainability and follow-up evaluation. Although the MMAT appraisal reflected moderately strong methodological quality, which suggests that solid methods were utilized, the intervention has limited applicability due to over-reliance on self-reported data.

The least effective program, according to our analysis, is the Stress Release Program [58] conducted in an Australian university, which was limited by the high attrition reported in this study, as well as the lack of adherence reporting and lack of long-term data.

#### 3.1.2. Evaluation of Intervention Development

It was found that interventions that were developed iteratively, and those with strong institutional backing and theoretical underpinning demonstrated higher feasibility than those with limited stakeholder engagement and poor adaptation strategies. Table 4 summarizes the key findings from the GUIDED assessment.

All three studies reporting on intervention development were well-designed and utilized structured approaches. Both the ARK program [62] and Booster Breaks program [63] incorporated flexibility and institutional tailoring, which may lead to higher feasibility of the programs. The Booster Breaks program [63] lacked long-term planning, which impedes on the tracking that can be performed using this intervention. Identified weaknesses of the WellCats program [61] included a lack of theoretical grounding and unclear integration of stakeholder input, which may hinder future feasibility by limiting the program’s replicability and conceptual coherence.

#### 3.1.3. Narrative Synthesis of Findings Across Both Analyses

Several factors determine an intervention’s effectiveness, for example, the level of participant engagement and institutional integration [61,62,63]. Our review indicates that interventions that are structured and have significant institutional support lead to greater participant adherence and long-term feasibility [58,59] whereas those that were self-driven or voluntary often struggled with retention [60]. Programs that integrated clearly defined components and facilitation strategies led to greater adherence and sustainability [61]. Similarly, workplace wellness initiatives like Booster Breaks [63] and the ARK program [62] demonstrated strong development frameworks, yet their effectiveness will depend on how well they were integrated into institutional policies and whether long-term monitoring is established. Interventions that were highly dependent on voluntary participation and self-motivation (such as the Stress Release Program and Booster Breaks) had high attrition rates due to competing demands [58,63].

The MMAT assessment showed that studies varied in methodological rigor. The use of validated measures was inconsistent across studies. Studies that were rated strongly showed clearer evidence for intervention outcomes than those with weaker methodological reporting.

#### 3.1.4. Facilitators of Intervention Success

The importance of institutional support and leadership buy-in emerged as one of the most critical facilitators in increasing participant engagement and ensuring long-term viability of interventions [61,62,63]. In addition, programs like the WellCats program, the ARK program, and the yoga-based intervention [59,61,62] that are deeply entrenched within the university structures benefit from being aligned with existing policies and workplace training encourages participation. Interventions that have included a clear schedule and defined components within a structured design had better adherence and retention [58,63]. The community therapy model [60] and the Booster Breaks buddy system [63] included elements of social support, which also allowed for increased engagement as participants foster group cohesion and had accountability built into the intervention.

Another key facilitator was that of flexibility; interventions that allowed for asynchronous delivery and participation, those that were self-paced, and those that allowed for workplace accommodations were more likely to succeed in settings as demanding as those in academic environments [59,60]. Overall, this emphasizes the need to design mental health interventions in the academic space that align with organizational priorities, promote community engagement, and minimize disruption to daily work responsibilities and deadlines.

#### 3.1.5. Barriers to Intervention Success

Despite these facilitators, workplace-based interventions faced persistent barriers that limited their impact and scalability. The most common barriers discussed across all included studies were those of lack of time and constraining workloads [57,58,59,60,61,62,63]. Across the seven studies (n ≈ 15,818), it was noted that many staff struggled to engage with mental health interventions during their normal working hours due to professional commitments. However, offering interventions outside of working hours did not necessarily resolve this issue. Participant-driven engagement outside of working hours was also limited in at least two studies in the United States and Australia [58,59], highlighting the ongoing challenge of scheduling and accessibility in intervention design. High attrition rates served as a further barrier in programs such as the Booster Breaks program [63] and the Stress Release Program [58].

The ability to assess the sustained impact of interventions was constrained by the fact that most lacked long-term follow-up evaluations of effectiveness. This is particularly true of the ARK [62] and the WellCats [61] programs, as process evaluations were not completed, making it difficult to determine how the interventions were implemented or maintained over time. The online delivery mode for the community therapy-based intervention [60] presented unique challenges associated with the use of technology; participants either struggled with access to the intervention or with digital literacy.

## 4. Discussion

The aim of this review was to evaluate the effectiveness of workplace-based mental health interventions aimed at academic staff and to identify key barriers and facilitators to their implementation. A variety of interventions were identified, with some incorporating organizational elements (e.g., leadership involvement or policy alignment) to enhance implementation. A combination of strong theoretical frameworks, structured implementation, institutional support, and peer engagement led to higher feasibility and better long-term adherence. Those interventions that were reliant on self-directed participation and lacked follow-up monitoring and evaluation, or those that did not receive consistent leadership support, faced greater challenges in achieving sustained impact. Our findings suggest that in addition to being methodologically sound, interventions that are woven into institutional policies, allowing for flexible delivery, and that are designed with long-term sustainability in mind could be the most effective.

Our review has found that structured and theoretically grounded interventions show better feasibility and engagement. Multiple studies [42,64,65,66] emphasize the importance of using theoretical frameworks, structured delivery, and evidence-based components to enhance the impact of mental health interventions. These studies also note that interventions grounded in theory are more adaptable, better received, and more sustainable across different institutional contexts.

The ARK intervention [62] and the yoga-based intervention [59] both included robust theoretical frameworks and, as such, will probably be easier to adapt for different academic workplace settings in the Global North or South. Theoretically grounded interventions result in complex interventions that are scalable and adaptable [67], making it possible to tailor them to new contexts and increasing the likelihood of successful adoption [68].

Another important finding from this review is that shorter individualized interventions appear to be more sustainable, especially in academic settings. Academic staff are often facing high workloads and competing interests as they have teaching responsibilities, take active roles in research and scholarship, and engage in service delivery, administration, and other duties. These intense and wide-ranging commitments appear to make it challenging for academic staff to engage in interventions. A recent similar systematic review on workplace-based mental health interventions in healthcare environments [69] found that brief interventions were the most effective type of intervention in the healthcare settings included in their review. Similarly, another review [70], also focused on healthcare workers, showed that shorter interventions can be as effective as longer-term interventions in reducing occupational stress. Whilst both of these reviews focused on healthcare workers, it is reasonable to expect that these findings could apply to academic settings, given the similar issues of workload intensity and time challenges. To support this assertion, the systematic review that focused on academic staff [42] showed that when combined with behavioral change elements and clear delivery plans, shorter interventions were the most effective interventions for academics [42]. Further research specifically on short interventions (and their long-term impact) in the academic space would strengthen our assumption.

While shorter interventions do show promise, their impact may be limited without broader structural changes. Some of the literature suggests that the improvements seen with individual-level interventions are not significant when compared to non-participants [71]. Another important factor to consider when assessing these types of interventions is that organizational-level changes would have a more substantial and long-term impact. Simply implementing individual-level interventions without addressing systemic pressures (such as the pressure to publish, performance metrics, and the marketization of universities) may result in limited improvements in academic staff well-being [72,73,74]. Overall, the literature supports the development of brief individualized interventions that are sustainable over time and embedded in an organizational structure that supports mental health and well-being for all. This also aligns with the United Nations Sustainable Development Goals, specifically SDG3: Good Health and Wellbeing and SDG16: Peace, Justice and Strong institutions [75].

It is difficult to estimate the long-term impact of interventions that lack longitudinal data and inconsistent follow-up evaluations. Sustainability remains a crucial consideration for workplace mental health interventions in any setting and is heavily dependent on institutional buy-in, active and methodical stakeholder engagement, and iterative program refinement as seen in the ARK program [62] and the ‘Booster breaks’, for example [63].

Crucial to any brief intervention is the manner in which it is implemented. We identified several facilitators contributing to successful intervention implementation. These can be categorized into three key areas: institutional support and leadership buy-in, integration within university structures, and social/work accommodations. Our findings that strong leadership commitment and institutional support facilitate the normalization of mental health promotion are strongly aligned with the evidence [65,76,77,78]. Beyond endorsement, sustainability also requires universities to provide structured support and dedicated funding. Without this, interventions risk losing momentum over time, as inconsistent leadership backing was shown to limit effectiveness across several studies.

Interventions that are embedded in university structures were more sustainable and impactful. Ways of undertaking this include adopting proactive and holistic strategies that focus on all stages of mental health: prevention, intervention, and accommodation [22,65]. Implementing efforts such as job stress reduction and enhancing job resources will lead to a shift in the organizational culture, which is imperative to ensure that interventions can be meaningful and effective [69]. These findings can be better understood through the lens of the World Health Organization’s guidance [3], which advocates for organizational interventions targeted at work environments that promote workplace culture and offer support throughout the employment cycle. Similarly, Reist et al. [79] emphasize the importance of integrated interventions to enhance access and outcomes.

In line with our review findings, the evidence indicates that interventions that included peer support and work accommodations (e.g., flexible hours, mental health leave, or adjusted teaching loads) enhance engagement and created psychologically safe spaces, which reflects best practice for inclusivity and psychological safety [79,80]. These work accommodations are especially helpful for those individuals who have a mental illness as they effectively contribute to a more psychologically safe environment for employees [81].

This could be due to a lack of adherence tracking and perceived burdens of continued participation in addition to the workload constraints indicated above. The possibility that stigma on campus might have had an impact is unexplored in any of the seven studies included. This includes both perceived external stigma (from colleagues or leadership) and internalized stigma that may deter staff from help-seeking or engaging in mental health interventions.

Our review identifies the following barriers to successful implementation or engagement with interventions in academic settings: lack of leadership/institutional support, workload pressures, high attrition rate from interventions, and technological challenges in digitally delivered interventions. These barriers align with the findings from a recent systematic review [78] that examined factors affecting the implementation of mental health interventions in the workplace in small-to-medium enterprises (SMEs). They found that participants did not engage with mental health initiatives that lacked managerial support, nor were participants willing to engage in these interventions due to high workloads and limited resources. They also found that competing interests often overshadowed the need to prioritize mental health interventions. Similarly, both factors (lack of prioritization by managers and high workloads/competing priorities) were also identified as barriers to effective workplace-based mental health interventions in a meta-synthesis review [82], further supporting our findings.

This pattern is also consistent with the USA-based Substance Abuse and Mental Health Services Administration (SAMHSA), a globally recognized authority on implementation science and mental health service delivery [83]. The SAMHSA emphasizes the importance of stakeholder buy-in from various levels (management and staff) for successful implementation of interventions. The absence of such support would be a significant barrier, as identified in our review. Additionally, the SAMHSA also highlights excessive workloads and inadequate staffing as significant obstacles to implementation; these barriers were also evident in academic settings in our review, where limited resources and competing priorities overshadow intervention planning and engagement.

The only study to use a digitally delivered intervention [60] reported connectivity issues, engagement challenges, usability concerns, and lack of technical support as barriers to implementation/uptake. Most of these are inherent in digitally delivered mental health interventions as digital literacy, privacy concerns, and platform accessibility often contribute to reduced uptake and effectiveness [84,85,86]. In addition to these well-established challenges, the literature has further identified time constraints and the need to be culturally sensitive as potential confounders to the success of interventions [87]. Furthermore, digital interventions and, in particular, mobile interventions are also prone to high attrition rates, which limit long-term effectiveness [88,89]. Potential solutions to minimize this include a user-centered approach, regular reminders, personalization of content, and providing incentives for participation [89]. These strategies could enhance engagement and improve retention rates, strengthening the evidence base of an increasingly digital work force.

This review is particularly relevant as it covers the COVID-19 pandemic period, during which academic staff faced significant challenges due to the momentous shift to online learning, balancing remote working, and increased isolation [90,91,92]. Our findings provide valuable insights into the feasibility and realities of implementing such interventions within the academic space; however, their broader impact and transferability are impacted by the studies’ methodological strengths and limitations.

Our assessment suggests that interventions that include strong methodological design, validated measures, and structured implementation processes [59,60,62] are the most reliable evidence for effectiveness. The broader literature also highlights the importance of methodological rigor in establishing confidence in intervention outcomes [93,94].

### 4.1. Methodological Strengths of the Review and Included Studies

This section outlines the methodological strengths identified both in our review process and in the included studies, as assessed through the GUIDED-led appraisal. This review was designed to generate robust conclusions and identify evidence gaps with greater precision. Understanding the barriers and facilitators of existing interventions may improve mental health intervention development.

Most of the studies showed strong coherence between the research objectives, methods, and analysis. As is expected from randomized controlled trials, there was higher methodological rigor in the intervention reported by Rodríguez-Jiménez et al., 2022 [59]. A similar review on mental health screening in the workplace critiqued the lack of high-quality randomized trials in this field, limiting our ability to draw reliable conclusions. Sena et al. (2023) [60] ensured that their findings were connected to participants’ lived experiences by adhering to strict phenomenological research principles. The depth of findings by Rodríguez-Jiménez et al. [59] is enhanced since they effectively integrated quantitative and qualitative approaches.

Considering the complexity of mental health interventions, our review incorporated diverse evidence types, to capture the nuanced experiences of academic staff. Alongside quantitative research, we included qualitative and mixed methods studies, providing deeper insights into the effectiveness and implementation of interventions. Triangulating findings from diverse approaches has enhanced understanding of the barriers and facilitators of existing workplace mental health interventions, strengthening the validity of our findings. This comprehensive approach helps inform the development of evidence-based, tailored strategies to support mental health for academic staff in HEIs.

Theoretically grounded interventions [62,63] provided a structured approach for assessing long-term impact, reinforcing their relevance to mental health research. This is also emphasized by the findings of the GUIDED evaluation; interventions anchored in strong theoretical models show greater feasibility and scalability. For example, a structured evaluation process facilitated adaptation of the ARK program across a variety of university settings, while the sustainability of the WellCats program was limited due to poor institutional integration and broad focus on wellness. Thus, robust foundations to workplace-based well-being interventions are critical, along with ensuring that these interventions are designed systematically, can be modified to suit varying audiences, and are sustainable.

### 4.2. Methodological Limitations of the Review and Included Studies

This section outlines the methodological limitations identified both in our review process and in the included studies, as assessed through the GUIDED-led appraisal. It is concerning that across the breadth of our searching, we identified only seven empirical studies written in English language that met our inclusion criteria. The lack of studies conducted in the Global South is saddening since many of these countries, often LMICs, have emerging and not yet mature institutes of higher education. This is an aspect that needs full attention moving forward.

The fact that only two out of the seven studies utilized control group methodologies is concerning. Conclusions around the effectiveness of those studies, which did not include a control group, are limited as we cannot make causal inferences in these instances. Similarly, two studies [57,60] relied on post-intervention assessments without comparators; the presence of a potential confounder thus implied less robust evidence. Similarly, the mindfulness intervention [58] did not adequately address the impact of confounders in their research design, meaning that any improvements in mental health status in this study cannot be fully attributed to the intervention. This study is also limited by a high attrition rate, which impacts on the completeness and reliability of their findings. Several other studies failed to monitor adherence rates, casting doubts on whether participants fully engaged with the program. There are also retention interventions to improve participant retention on RCTs and these also need to be grounded on sound theoretical frameworks and participant input [95].

The small sample sizes across most studies, together with the inclusion of homogeneous study populations (e.g., the NEWSTART intervention was conducted exclusively in Seventh-Day Adventist institutions, and the hatha yoga/body–mind awareness program only had thirty-one participants), were a key limitation identified in this review. These small sample sizes restrict the generalizability of the findings. Only one study [59] tested and reported statistical significance. For the remaining studies, there is limited confidence in terms of whether their results can be applied to broader academic staff populations. The low statistical power has a significant negative impact on the broader applicability of these study findings.

Due to the lack of long-term follow-up data across most studies, it is not easy to determine the long-term sustainability of these interventions. This is not unique, as other reports emphasize the need for longitudinal studies in order to strengthen confidence in intervention effectiveness [64,65,66]. Similar critiques exist from those working in the field of mental health and academic staff well-being reporting poor tracking and the absence of long-term follow-up [43,44]. Our recommendations regarding stakeholder feedback inclusion and assessment from both an individual and organizational perspective align closely with Luu et al., 2024 [40], who advocate for longitudinal mixed method approaches in future research.

We strongly advocate for workplace-based mental health interventions to be tailored to academic settings, considering the unique challenges and opportunities that are inherent in these spaces. This approach is also supported by other authors in the field [65], who also advocate for the role of comprehensive stakeholder engagement.

### 4.3. Limitations of This Review

A critical limitation of this review is to discuss the geographic bias of having six out of the seven included studies based in the Global North, as this has significant impact on intervention transferability. Contextual factors such as socio-culture and local interpretations regarding mental health vary significantly between the two regions, and effective mental health interventions need to be culturally sensitive [96,97]. Universality of approaches is often assumed when applying interventions emerging from high-income settings to LMICs, which can often lead to ineffective or culturally inappropriate interventions in the Global South [97]. Similarly, disparities in knowledge and resource allocation between the Global North and South further add to this divide. Broader determinants of mental health such as economic stability and community support, which mostly apply to the Global South, may be undermined by studies in the Global North in favor of biomedical interventions; this perpetuates already established concerns such as epistemic justice and medical imperialism [96]. The predominance of studies conducted in the Global North limits insights into the interventions’ applicability in different cultural and institutional contexts, particularly resource-constrained settings often seen in the Global South [98]. This geographic concentration not only reflects where research has been conducted but also which types of papers are accessible through English language databases. This review only included studies published in English and did not include non-English databases such as the Latin American and Caribbean Literature in Health Sciences or China National Knowledge Infrastructure databases, which may have excluded relevant evidence from underrepresented regions. Future research and reviews should explicitly examine interventions adapted for diverse geographical and linguistic contexts to improve the transferability and global relevance of findings. Finally, the search was completed in January 2024; therefore, more recent publications were not captured and should be incorporated in future updates or reviews to maintain currency.

Overall, this review emphasizes the need to design mental health interventions in the academic space that align with organizational priorities, promote community engagement, and minimize disruption to daily work responsibilities. Contextual adaptation needs to be a critical factor in designing and implementing mental health interventions for academics, including their institutions and national contexts as well. Leadership styles, staff demographics, and availability of resources will ultimately influence the feasibility and effectiveness of these interventions [97,99], but should be considered and offered to facilitate staff health and well-being. It is also imperative that these interventions be adapted around cultural norms, take stakeholder input into account, and be embedded into institutional structures such as HR policies and wellness frameworks. Finally, due to the ever-present competing interests that plague academics, interventions targeted at this group must be operationally feasible, considering the realities of academic work.

## 5. Conclusions

This review systematically identified effective university-based mental health interventions for academic staff and explored potential facilitators and barriers to the implementation of these interventions. Flexible, institutionally supported, and theoretically grounded interventions, particularly those that are brief and address workload-related issues, show the potential for engagement and long-term sustainability. Those interventions that lacked follow-up mechanisms were more likely to be ineffective. Unfortunately, the evidence base of this review was constrained by methodological weaknesses across the seven included studies (e.g., small sample sizes, lack of control groups) coupled together with the predominance of findings based in Global North settings.

Notably, our review highlights the limited volume of peer-reviewed research evaluating mental health interventions specifically for academic staff. This may be due to a general under-prioritization of intervention work in this population. Thus, we re-emphasize the critical importance of developing and evaluating culturally relevant interventions that are context-specific as well as the need for longitudinal studies that assess long-term outcomes and scalability. Finally, interventions that are co-designed, with and for staff, could support more meaningful uptake and longer-term sustainability within HEIs. More broadly, our findings reinforce the importance of workplace health promotion and safety strategies as integral components of sustainable occupational health practice.

## Figures and Tables

**Figure 1 ijerph-22-01787-f001:**
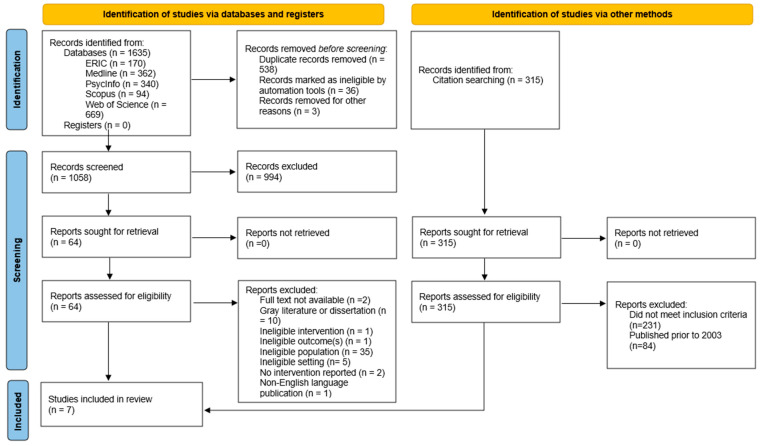
PRISMA flowchart of study selection.

**Table 1 ijerph-22-01787-t001:** Study characteristics of included studies.

Citation	Country	Study Design	Control	Sample	Intervention Summary	Outcome (Measure)	Follow-up	Institutional Support	Key Findings
Category 1: Studies focusing on intervention effectiveness									
Ashley & Cort (2007) [57]	United States	Quantitative non-randomized	No	Faculty at Seventh Day Adventist institutions (*n* = 124)	NEWSTART: Self-guided lifestyle program with an emphasis on nutrition, water, exercise, sunshine, temperance, and trust in divine power.	Faculty stress (FSI)	No	No	Reduced stress with limited generalizability of findings.
Koncz et al. (2016) [58]	Australia	Quantitative non-randomized	Yes	University staff Baseline: INT: *n* = 71 CONT: *n* = 80 Post-intervention: INT: *n* = 45 CONT: *n* = 39	Stress Release Program: Mindfulness-based stress reduction initiative. In-person group sessions for 7 weeks.	Psychological distress (GHQ-12)	No	University-endorsed initiative	Significant reductions in psychological distress and work engagement.
Rodríguez-Jiménez et al. (2022) [59]	Spain	Quantitative randomized controlled pre–post	Yes	University teachers Hatha yoga: *n* = 11 BMA: *n* = 10 CONT: *n* = 10	Research-based stress reduction incorporating hatha yoga and body–mind awareness (BMA). In-person 90 min weekly sessions for 8 weeks.	Self-reported well-being and stress	No	University collaboration	Hatha yoga reduced stress; BMA improved self-awareness and communication.
Sena et al. (2023) [60]	Brazil	Qualitative	No	Professors in nursing (*n* = 8)	Integrative Community Therapy (ICT): Open support circles for stress and resilience. Twelve weekly sessions.	Perceptions of well-being	No	No	ICT circles provided support for mental health, especially during COVID-19.
Category 2: Studies focusing on intervention development									
Lloyd et al. (2017) [61]	United States	Mixed methods	N/A	University staff (*n* = NR)	WellCats: University-based initiative for promoting health and well-being.	Institutional well-being	Long-term evaluation planned over 16 months	Embedded within institutional policies	Promising intervention design; lacked detail about involvement of stakeholders.
Innstrand & Christensen (2018) [62]	Norway	Mixed methods	N/A	University staff across multiple institutions (*n* = 15,000+ responses collected)	ARK: Holistic intervention for workplace mental health and well-being, driven by survey data.	Psychosocial work environment and staff well-being	Every 2–3 years	Multi-university collaboration with leader involvement	Theory-driven and well-structured with long-term adaptability.
Taylor et al. (2020) [63]	United States	Mixed methods	N/A	University staff (*n* = NR)	Logic Model Approach to Stress Management: A structured, facilitated goal-setting intervention using a logic model.	Workplace stress and work engagement	Short-term follow-up after intervention	Institutional framework for mental health at work	Clear logic model-based approach; limited transparency in terms of methodology.

**Table 2 ijerph-22-01787-t002:** Summary of theoretical underpinnings and mental health outcomes of included studies.

Study	Theoretical Underpinnings	Mental Health Outcomes Assessed
Ashley & Cort, 2007 [57]	Transactional Model of Stress and Coping; Lifestyle Medicine (Seventh-Day Adventist health principles)	Stress reduction; self-efficacy; life satisfaction
Innstrand and Christensen, 2018 [62]	Job Demands–Resources (JD-R) Model; participatory decision-making; resource-focused intervention planning	Workplace engagement; burnout prevention; psychosocial well-being
Koncz et al., 2016 [58]	Mindfulness principles; cognitive reframing; breathwork (no explicit framework named)	Stress reduction; psychological distress; workplace well-being
Lloyd et al., 2017 [61]	Social Cognitive Theory; Transtheoretical Model; Self-Determination Theory; Ecological Model	Workplace well-being; health behavior change; long-term engagement
Rodriguez-Jiminez et al., 2022 [59]	Body awareness and mindfulness principles; movement-based stress reduction (no explicit framework named)	Stress reduction; subjective well-being; self-awareness
Sena et al., 2023 [60]	Social and Group Psychology Theories; peer-support frameworks; collective problem-solving	Emotional resilience; workplace well-being; social connectedness
Taylor et al., 2020 [63]	Logic Model Approach; goal-setting and structured implementation for stress management	Workplace stress reduction; employee engagement; physical activity

**Table 3 ijerph-22-01787-t003:** MMAT quality appraisal summary.

Author(s), Year	MMAT	Key Strengths	Key Limitations
Ashley & Cort, 2007 [57] (lifestyle-based intervention)	80%	Articulated objectives, validated measures (Faculty Stress Index)	Limited representativeness, potential confounders
Koncz et al., 2016 [58] (mindfulness-based intervention)	60%	Clear intervention description, validated outcome measures, appropriate measurement tools (GHQ-12)	High attrition, incomplete follow-up, and confounder risks
Rodriguez-Jiminez et al., 2022 [59] (yoga-based intervention)	100%	Robust randomization, clear integration of methods, validated outcome measures	Limited generalizability due to small sample size
Sena et al., 2023 [60] (community-based therapy)	100%	Explicit methodology, clear analytical and theoretical grounding	Small sample limits broader applicability

**Table 4 ijerph-22-01787-t004:** Mapping of intervention development studies to the GUIDED framework.

Citation	Intervention Summary	Adherence to GUIDED	Strengths	Limitations
Lloyd et al., 2017 [61]	WellCats Program: Institutional wellness program in a U.S. university	Moderate	Clear institutional context, structured approach	Limited stakeholder transparency
Innstrand & Christensen, 2018 [62]	ARK intervention: A systematic, bottom-up program to improve university psychosocial work environments and employee well-being.	High	Comprehensive stakeholder engagement, iterative refinement, and strong theoretical underpinning	No significant limitations identified
Taylor et al., 2020 [63]	Booster Break Program: Large university medical center, USA, targeting workplace stress reduction	Moderate	Well-defined logic model, structured implementation	Limited transparency in stakeholder involvement

## Data Availability

All data supporting the findings of this study are contained within the article and its Appendix A. No new datasets were generated or analyzed during the course of this study.

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
