# Peer review of "Effectiveness and Feasibility of Workplace-Based Mental Health Interventions for University Academic Staff: A Systematic Review"

_ijerph, 2025, doi:10.3390/ijerph22121787_

Round 1
Reviewer 1 Report
Comments and Suggestions for Authors
Thank you,
1. Make a clear connection between the research questions and the gaps.
Author/s mentioned that there isn't much research on interventions for academic personnel, where is the evidence for this statement? I suggest the author/s could be more clear:
"Even though there is more interest in mental health at work, systematic reviews have not yet fully put together evidence on the effectiveness, barriers, and facilitators of mental well-being interventions for academic staff." This review fills this gap by...
2. Author/s should support the necessity of a systematic review as opposed to a scoping or narrative review.
Example: In conclusion, briefly explain why systematic review approach is the best suitable:
3. I suggest author/s should bring stronger link between general workplace initiatives and academic staff.
4. I recommend make the focus on significances richer. The study questions emphasize effectiveness and barriers/facilitators; however, the introduction should also address critical outcomes such as burnout, stress, depression, and productivity to delineate the scope and significance of the review.
Author Response
Please find the author's responses to the reviewer's comments in the attachment.

Reviewer 2 Report
Comments and Suggestions for Authors
I have completed the revisions of the manuscript. Overall, the manuscript is in good; however, there are a few points that needed minor corrections and clarifications. You can find my notes and explanations regarding these points in the file I have attached. Please feel free to review the document at your convenience. Thank you.

Author Response

(The authors gave the same response as above.)

Reviewer 3 Report
Comments and Suggestions for Authors
Overall, this article's content is not particularly novel, as it aligns with other systematic reviews on workplace mental health interventions and the broader literature on the topic. However, the decision to specifically address academic staff as the target population is noteworthy and has clear practical implications. The surprisingly small number of selected studies suggests how underresearched the topic is and supports the need for more research in this area. The study is well designed and conducted; the methodological procedures described appear rigorous; and the manuscript is well structured and organized. I suggest the authors make a few minor revisions before publication.
- Throughout the manuscript, the study is referred to as a “mixed-studies systematic review”. This could give the erroneous impression that all the selected studies were mixed-method studies, which was not a selection criterion. I suggest correcting this to avoid misinterpretation.
- The rationale for selecting the timespan of 2003-2023 is unclear. Studies from before 2003 and after 2023 could offer valuable insights. The authors should explain their reasoning. One option would be to integrate the results with studies from those missing years.
- “While the review predominantly focused on interventions oriented towards individual-level action, programs operating at both individual and organisational levels (including those that describe intervention development) were included, where relevant”. This criterion is unclear. The authors should clarify whether they were looking for individual- or organizational-level interventions, or both, as per the selection criteria. They should also define what they mean by “individual” and “organizational” to clarify how the studies were selected against the established definitions.
Author Response

(The authors gave the same response as above.)
